# Hydroxyethylcellulose-Based Hydrogels Containing Liposomes Functionalized with Cell-Penetrating Peptides for Nasal Delivery of Insulin in the Treatment of Diabetes

**DOI:** 10.3390/pharmaceutics14112492

**Published:** 2022-11-17

**Authors:** Eliete de Souza Von Zuben, Josimar Oliveira Eloy, Maiara Destro Inácio, Victor Hugo Sousa Araujo, Amanda Martins Baviera, Maria Palmira Daflon Gremião, Marlus Chorilli

**Affiliations:** 1Department of Drugs and Medicines, School of Pharmaceutical Sciences, São Paulo State University (UNESP), Rodovia Araraquara-Jaú, Km1, Araraquara 14800-903, Brazil; 2Department of Pharmacy, School of Pharmacy, Dentistry and Nursing, Federal University of Ceará, Fortaleza 60028-181, Brazil; 3Department of Clinical Analysis, School of Pharmaceutical Sciences, São Paulo State University (UNESP), Rodovia Araraquara-Jaú, Km1, Araraquara 14800-903, Brazil

**Keywords:** insulin, liposome, cell-penetrating peptide, hydrogel, nasal delivery, antihyperglycemic effect

## Abstract

Liposomes functionalized with cell-penetrating peptides are a promising strategy to deliver insulin through the nasal route. A hydrogel based on hydroxyethylcellulose (HEC) aqueous solution was prepared, followed by a subsequent addition of liposomes containing insulin solution functionalized with trans-activator of transcription protein of HIV-1 (TAT) or Penetratin (PNT). The formulations were characterized for rheological behavior, mucoadhesion, syringeability, in vitro release and in vivo efficacy. Rheological tests revealed non-Newtonian fluids with pseudoplastic behavior, and the incorporation of liposomes (HLI, HLI_TAT_ and HLI_PNT_) in hydrogels did not alter the behavior original pseudoplastic characteristic of the HEC hydrogel. Pseudoplastic flow behavior is a desirable property for formulations intended for the administration of drugs via the nasal route. The results of syringeability and mucoadhesive strength from HEC hydrogels suggest a viable vehicle for nasal delivery. Comparing the insulin release profile, it is observed that HI was the system that released the greatest amount while the liposomal gel promoted greater drug retention, since the liposomal system provides an extra barrier for the release through the hydrogel. Additionally, it is observed that both peptides tested had an impact on the insulin release profile, promoting a slower release, due to complexation with insulin. The in vitro release kinetics of insulin from all formulations followed Weibull’s mathematical model, reaching approximately 90% of release in the formulation prepared with HEC-based hydrogels. Serum insulin levels and the antihyperglycemic effects suggested that formulations HI and HLI have potential as carriers for insulin delivery by the nasal pathway, a profile not observed when insulin was administered by subcutaneous injection or by the nasal route in saline. Furthermore, formulations functionalized with TAT and PNT can be considered promoters of late and early absorption, respectively.

## 1. Introduction

Worldwide, diabetes mellitus (DM) has been reported as a public health problem, as it affects individuals of all ages, including children, adults and elderly people, in addition to pregnant women [1,2,3]. According to the International Diabetes Federation (IDF), there were 537 million people with diabetes in 2021, and this number should reach 643 million by 2030 and 783 million by 2045. Currently, the number of people with undiagnosed diabetes (particularly type 2 diabetes) exceeds 50%. In 2021, approximately 6.7 million deaths worldwide were attributed to DM. The global public health expenditure for the treatment of people with DM was estimated at USD 966 billion in 2021, projected to reach USD 1028 billion in 2030 and USD 1054 billion in 2045 [4,5].

Many patients with DM depend on subcutaneous administration of insulin to achieve effective glycemic control, as well as to prevent the development of several complications of this syndrome. Nevertheless, it is observed that this form of invasive insulin administration becomes restricted, in addition to being associated with some discomfort, such as pain from daily applications and local infection, which can contribute to the decrease in patient adherence to the treatment. The search for an alternative therapy has prompted a variety of studies; among these is the development of new insulin release systems that permit the use of alternative routes to parenteral. An example is the inhalable drug Afrezza^®^ (human insulin), a inhalation insulin powder for patients with diabetes requiring prandial insulin [6,7,8].

Among non-invasive drug delivery systems, the nasal route has been the subject of several studies on alternative routes for the administration of systemically active drugs, such as proteins that are widely metabolized by the gastrointestinal tract and/or poorly absorbed orally. This route offers painless administration, besides being well suited for drug absorption, since it has a large area of epithelial surface available due to numerous microvilli and has a highly vascularized subepithelial layer [9,10]. The disadvantages of this route include the very active mucociliary clearance, the possibility of enzymatic degradation since the nasal cavity contains different types of enzymes, and the low permeability of the nasal mucosa that often results in low bioavailability. To develop an efficient nasal insulin delivery system, it is necessary to overcome these barriers. For this purpose, some formulation strategies can be employed for the controlled delivery of insulin via the nasal route. Herein, we address the use of hydrogels containing insulin-loaded liposomes functionalized with cell-penetrating peptides (CPPs), such as TAT and PNT, that can penetrate cells and thus are promising for therapeutic purposes [11,12,13,14].

Liposomes are small artificial vesicles of spherical shape, which vary in size from the nanometer scale to a few micrometers composed of phospholipid bilayers surrounding an aqueous inner compartment, which allow the encapsulation of hydrophilic and lipophilic drugs, thus constituting relevant systems for the transport, encapsulation and sustained release of drugs. Moreover, liposomes are biocompatible and biodegradable, allowing for prolonged drug release, decreasing toxicity, increasing bioavailability and reducing collateral effects, and within this context, insulin-loaded liposomes may be found in the literature [15,16,17,18,19,20]. Several findings in the literature demonstrate the use of liposomes for drug delivery [21,22,23,24].

To enhance the nasal delivery of insulin, in this study, we explored the potential of cell-penetrating peptides (CPPs) or protein transduction domains (PTDs), which are a class of diverse peptides with different chemical structures, typically with 5–30 amino acids with positive charge. CPPs can permeate cells through the cell membrane and promote the uptake of a variety of non-covalently or covalently linked cargoes, including proteins, DNA, siRNAs, antibodies, oligonucleotides, small drugs and different nanocarriers such as nanospheres, nanocapsules and liposomes [25,26,27]. The cell internalization mechanisms are direct translocation and endocytosis [28,29]. The discovery of the first cell-penetrating peptide in 1988, the cationic peptide TAT (GRKKRRQRRRPPQ), derived from the HIV-1 TAT protein, was followed by the identification in 1991 of PNT (RQIKIWFQNRRMKWKK), derived from Antennapedia, a homeoprotein of Drosophila melanogaster. Both peptides can be internalized by cells, with very limited toxicity, crossing cell membranes [30,31,32,33]. Additionally, in a previous study, Khafagy and collaborators investigated the influence of CPPs (L-PNT or D-PNT) and (L-R8 or D-R8) on insulin nasal delivery, but did not investigate the role of liposomes nor hydrogels; therefore, the strategy employed herein is new [34].

Hydrogels of hydroxyethylcellulose (HEC) are an example of nonionic, biocompatible, water-soluble polymers and one of the most important commercially available cellulose derivatives, useful for topical formulations. They are widely soluble in cold or hot water and are used in the preparation of formulations with a wide range of viscosities. They are commonly employed in the preparation of solid, liquid and semi-solid formulations and controlled-release drug delivery systems [35,36,37,38,39]. However, the main goal of this study was to investigate the pharmaceutical performance of HEC-based hydrogel as a viscous vehicle, due to the few original publications addressing its influence on nasal delivery of insulin, combined or not with liposomes and CPPs.

Thereby, we aimed to develop and characterize HEC-based hydrogels containing insulin-loaded liposomes functionalized with cell-penetrating peptides (TAT and PNT) for intranasal administration. The in vivo efficacy of this nasal insulin delivery system was investigated by analyzing the insulin serum levels in non-diabetic rats, as well as the antihyperglycemic responses and the tissue insulin signaling activation in streptozotocin-diabetic rats.

## 2. Materials and Methods

### 2.1. Materials

The human insulin solution Novolin R^®^ (100 IU mL^−1^, Lot GS63F52) was from Novo Nordisk, Denmark. Cholesterol (CH) was obtained from Sigma-Aldrich, USA (purity ≥99%). The cell-penetrating peptides (CPPs), TAT and PNT were used according to the manufacturer’s instructions and obtained from Aminotech^®^ (Brazil). Chloroform was supplied by Merck, USA. Soybean phosphotidylcholine (PC) was obtained from Lipoid GMBH, Germany. HEC (HEC, Natrosol 250 HHX) was purchased from Synth. Mucin from porcine stomach was from Sigma-Aldrich (Steinheim, Germany). Ultra-purified water was obtained by a Milli-Q system (Millipore, Burlington, MA, USA) and used for all the experiments. Streptozotocin (STZ) and the Rat/Mouse Insulin ELISA Kit (EZRMI-13K) were purchased from Merck and Sigma-Aldrich. Phosphate-buffered saline (PBS) is made of NaCl 137 mmol/L, KCl 2.6 mmol/L, Na_2_HPO_4_·12H_2_O 6.4 mmol/L and NaH_2_PO_4_ 1.4 mmol/L, pH 7.4. All other reagents used were of analytical grade.

### 2.2. Preparation of Insulin-Loaded Liposomes

Liposomes were prepared by the hydration of the thin lipid film as previously described in the literature [40,41,42]. Briefly, 100 mg of a phospholipid mixture, including cholesterol (CH) and soybean phosphatidylcholine (PC), was weighed (weight ratios of 3:5) and dissolved in a methanol/chloroform mixture (2:1 *v*/*v*) into a round-bottom flask attached to a rotary evaporator for 30 min at 40 °C. Then, the lipid film formed was hydrated for 30 min at 40 °C/100 rpm, with 10 mL of phosphate buffer (PBS, pH 7.4) containing 4 mL of insulin solution (1400 µg mL^−1^), obtaining the liposomal suspension, followed by high-pressure homogenization (Avestin, EmulsiFlex-C3, Canada, 10,000 psi). Liposomes with uniform size were obtained after the homogenization of liposome suspension [43].

### 2.3. Functionalization of Insulin-Loaded Liposomes with TAT and Penetratin

Specific amounts (1 mM) of cell-penetrating peptides (PNT and TAT) were weighed and dissolved in Milli-Q water. The CPP solutions and liposomes containing insulin solution (Novolin R^®^) prepared as previously described were mixed gently at room temperature and incubated for electrostatic interaction between negatively charged liposome and positively charged CPPs for 30 min, to obtain a final concentration of 0.1 mM for both PNT and TAT [34,44].

### 2.4. Preparation of Hydrogel

To prepare the hydrogel, in 10 mL of ultra-purified water at 40 °C, 200 mg of HEC was dissolved under constant magnetic stirring for 30 min, in order to obtain a non-ionic gel at 2% (*m*/*v*). Then, the solution was kept for 24 h in a refrigerator at 2–8 °C in a 15 mL Falcon^®^ tube, to ensure complete dissolution and homogeneity. Subsequently, in 5 mL of HEC hydrogel, 15 mL of liposomal dispersion containing insulin solution functionalized with CPPs was incorporated [45,46,47].

### 2.5. Rheological Behavior

All rheological measurements of the formulations containing HEC hydrogels (HEC-based hydrogel containing insulin, HI; HEC-based hydrogel containing insulin-loaded liposomes, HLI; HEC-based hydrogel containing insulin-loaded liposomes functionalized with PNT, HLI_PNT_; HEC-based hydrogel containing insulin-loaded liposomes functionalized with TAT, HLI_TAT_) were carried out in triplicate on rheometer HR-2 (TA Instruments) with a cone and plate geometry of 40 mm diameter (cone angle 2°) and gap of 52 µm. Aliquots of the formulations were allowed to equilibrate for at least 3 min prior to analysis and thoroughly applied to the lower plate of the equipment to avoid sample shearing.

#### 2.5.1. Flow Measurements

The flow rheology was studied to determine the flow properties of the formulations using a shear rate within the range of 0.1 to 100 s^−1^ (upward curve) and 100 to 0.1 s^−1^ (downward curve), each stage lasting 120 s, held at the upper limit for 10 s between the curves [48]. All measurements were made at 32 ± 0.2 °C and 37 ± 0.2 °C in triplicate and a fresh sample was loaded for each run. Modeling of the flow properties of each formulation was performed using the Power Law (Ostwald–de Waele equation), described in Equation (1) [49]:τ = κ · γ^n^(1)
where τ is the shear stress (Pa), κ is the consistency index (Pa.s), γ is the shear rate (s^−1^) and “n” is the flow behavior index.

#### 2.5.2. Oscillatory Measurements

Initially, oscillatory measurements were conducted to determine the viscoelastic region. For this analysis, a range of stress sweep from 0.01 to 10 Pa and a frequency of 1 Hz were used. Constant shear stress of 1 Pa was then selected on the linear viscoelastic region and the frequency sweep analysis was performed to determine the loss modulus (G″) and the storage modulus (G′). For this assay, the frequency range of 0.01 to 10 Hz at 32 ± 0.2 °C and 37 ± 0.2 °C was used, at a constant stress of 0.3 Pa. All measurements were performed in triplicate with fresh samples for each run [48].

### 2.6. Chromatographic Conditions for Insulin Quantification

For insulin quantification, high-performance liquid chromatography (HPLC) was employed (model 1200, Agilent Technologies LC system with DAD/UV–visible detector) as previously described by Von Zuben and collaborators [50]. An RP C18- Luna^®^ Phenomenex (4.6 mm × 250 mm, 5 μm particle size) column was used at room temperature (25 ± 1 °C). The mobile phase was composed of 0.1% TFA aqueous solution and acetonitrile (40:60, *v/v*) at an isocratic flow rate of 1.0 mL min^−1^. The detector was set to 214 nm and peak areas were integrated automatically (ChemStation). The calibration curve of insulin was prepared in PBS, pH 7.4, at concentrations from 0.5 to 100 µg mL^−1^. PVDF membrane filter from Millipore, USA, with 0.45 μm was employed to filter all the samples.

### 2.7. In Vitro Release of Insulin from Hydrogels

Franz’s cell apparatus (Microette Plus, Hanson Research, Chatsworth, EUA) was employed to determine in vitro release of insulin from hydrogel-based formulations. For the assay, Franz diffusion cells with an approximate volume of 7 mL were used. A synthetic cellulose acetate membrane with an area of 1.77 cm^2^ was hydrated with phosphate buffer (PBS, pH = 7.4) prior to analysis. Then, 300 μL of the hydrogel formulations with insulin-loaded liposomes functionalized with cell-penetrating peptides (HLI_TAT_ and HLI_PNT_) were studied. To simulate the nasal temperature, the assay was conducted at 32 ± 0.5 °C with constant stirring (300 rpm) of the receptor solution (PBS, pH = 7.4). The release sample of 1.5 mL was automatically collected from the receptor compartment in the following times: 5 min, 30 min, 1 h, 2 h, 4 h, 8 h, 12 h, 16 h and 24 h. The volumes were immediately replaced with fresh receptor solution. As previously described, the samples were quantified by the HPLC method and the assay was performed on six replicates for each sample [51,52]. The insulin release profiles were calculated from Equation (2):(2)Q=Ct·Vr+ Σ Vc·Cc
where Q (μg cm^2^) represents the total amount of insulin permeated up to time t, Ct (µg mL^−1^ cm^2^) is the insulin concentration measured at time t, Vr (mL) is the receptor solution volume (7 mL), Cc (μg mL^−1^ cm^2^) is the concentration at the previous sampling and Vc (mL) is the volume sampled.

The insulin release profiles were fitted to different mathematical kinetic models, such as the Weibull, First-Order, Baker–Londsdale, Korsmeyer–Peppas, Hixon–Crowell, Higuchi and Hopfenberg models and for better discussion of the release mechanism [53,54,55].

### 2.8. Experimental Design for In Vivo Studies

#### 2.8.1. Animals

Male Wistar rats (*Rattus norvegicus*) weighting 140–160 g (6 weeks old) were maintained in polypropylene cages under environmentally controlled conditions of temperature (23 ± 1 °C) and humidity (55 ± 5%) and a 12 h light/dark cycle, with free access to daily feedings of normal lab chow diet and water throughout the experiment. Thirty-five non-diabetic rats and sixty diabetic rats were used in these experiments. The Committee for Ethics in Animal Experimentation of the School of Pharmaceutical Sciences, Unesp, Araraquara, SP, Brazil (CEUA/FCF/CAr resolution number 04/2018) approved the experimental procedures.

#### 2.8.2. Temporal Analysis of the Insulin Serum Levels

Non-diabetic male Wistar rats were used to assess the insulin serum levels after the intranasal administration of the formulations. An animal random division was held in 7 groups (n = 5), as follows: (i) intranasal administration of saline (0.85% NaCl) (SS_nasal_); (ii) subcutaneous administration of 0.25 IU insulin (Novolin R^®^) (INSsub); (iii) intranasal administration of 2.0 IU insulin prepared in saline (INSnasal); (iv) intranasal administration of HEC-based hydrogel containing 2.0 IU insulin (HI); (v) intranasal administration of HEC-based hydrogels containing 2.0 IU insulin-loaded liposomes (HLI); (vi) intranasal administration of HEC-based hydrogels containing 2.0 IU insulin-loaded liposomes functionalized with TAT (HLI_TAT_); (vii) intranasal administration of HEC-based hydrogels containing 2.0 IU insulin-loaded liposomes functionalized with PNT (HLI_PNT_). The insulin dose administered subcutaneously was 0.25 IU/rat [56]. The total volume administered intranasally of saline containing insulin (2.0 IU insulin/formulation/rat) or formulations was 100 μL, divided into 25 μL applications introduced alternately in each nostril at a time, using a micropipette to instill [57].

The rats were fasted for 12 h and anesthetized by intraperitoneal injection (ketamine 45 mg kg^−1^ and xylazine 5 mg kg^−1^), and they were kept for 5 min in heated boxes for peripheral to induce vasodilation. Afterwards, blood samples (50 μL) were taken from the tip of the tail at described times: 0 (before intranasal or subcutaneous administration of the preparations), and 60, 120, 180, 240 and 360 min after the treatments. The blood was centrifuged for 10 min at 6400× *g*. The serum was separated and conditioned at −20 °C for the subsequent measurement of the insulin serum levels by using Rat/Mouse Insulin ELISA Kit (Merck S/A, EZRMI-13K). The results are described in ng mL^−1^.

At the end of the experiment, euthanasia was performed by deep anesthesia, according to the Brazilian Guide to Good Practices in Euthanasia in Animals—Recommended Concepts and Procedures, Federal Council of Veterinary Medicine 2012 and Normative Resolution 13/2013.

#### 2.8.3. Anti-Hyperglycemic Responses in Diabetic Rats

After an acclimation period, experimental type 1 diabetes mellitus was induced in previously 12 h fasted rats by a single intravenous injection of 40 mg/kg STZ prepared in 0.01 M citrate buffer (pH 4.5). Isoflurane inhalation was used to anesthetize all rats [58]. Three days after the STZ administration, rats with postprandial glycemia values ≥ 350 mg dL^−1^ were selected for the study. Glycemia levels were determined by using a glucometer (Abbott Diabetes Care Ltd., Brooklin, Brazil).

Rats were distributed into ten different groups (six rats per group) by matching specimens having similar values of glycemia and body weight, as follows: (i) intranasal administration of saline (0.85% NaCl) (SS_nasal_); (ii) subcutaneous administration of 2.0 IU insulin (Novolin R^®^) (INS_sub_); (iii) intranasal administration of 2.0 IU insulin prepared in saline (INS_nasal_); (iv) intranasal administration of HEC-based hydrogel containing 2.0 IU insulin (HI); (v) intranasal administration of HEC-based hydrogels containing 2.0 IU insulin-loaded liposomes (HLI); (vi) intranasal administration of HEC-based hydrogels containing 2.0 IU insulin-loaded liposomes functionalized with TAT (HLI_TAT_); (vii) intranasal administration of HEC-based hydrogels containing 2.0 IU insulin-loaded liposomes functionalized with PNT (HLI_PNT_); (viii); intranasal administration of HEC-based hydrogels containing blank liposome (HLB). The insulin dose administered subcutaneously was 2.0 IU/rat. The total volume administered intranasally of saline containing insulin (2.0 IU insulin/formulation/rat) or formulations was 100 μL, divided into 25 μL applications introduced alternately in each nostril at a time, using a micropipette to instill [57].

Seven days after diabetes mellitus installation, the rats were fasted for 12 h and anesthetized by intraperitoneal injection (ketamine 45 mg kg^−1^ and xylazine 5 mg kg^−1^), and they were kept for 5 min in heated boxes for peripheral to induce vasodilation. After that, blood samples (50 μL) were taken from the tip of the tail at described times: 0 (before intranasal or subcutaneous administration of the preparations) and 30, 60, 90, 120, 180, 240, 300 and 360 min after the treatments for monitoring blood glucose levels, using a glucometer (FreeStyle^®^ Optium Neo). The results are described in mg dL^−1^.

At the end of the experiment (360 min), diabetic rats were euthanized by carbon dioxide (CO_2_) via inhalation, and the tibialis anterior skeletal muscles were immediately removed and kept at −80 °C before analysis. Skeletal muscles were used for the analysis of the changes in the phosphorylation levels of AKT (protein kinase belonging to the insulin signaling cascade) at the Ser-473 residue.

##### Western Blotting Analysis

The tibialis anterior muscles were homogenized in Tris-HCl buffer (50 mM; pH 7.4) containing 1 mM EDTA, 150 mM NaCl, 24 mM sodium deoxycholate, 1% Triton X-100, 0.01% sodium dodecyl sulfate (SDS), protease inhibitors (1.5 µM aprotinin, 2.1 µM leupeptin, 1 mM phenylmethanesulfonyl fluoride) and phosphatase inhibitors (10 mM sodium pyrophosphate, 100 mM sodium fluoride, 10 mM sodium orthovanadate). After homogenate centrifugation (4 °C, 11,900× *g* for 30 min), protein concentration of the supernatant was determined [59]. Equal volumes of supernatant and sample buffer (10% glycerol, 62.5 mM Tris-HCl, 2% SDS, 33.2 mM dithiothreitol (DTT), 0.01% bromophenol blue, pH 6.8) containing 100 μg of protein were subjected to SDS-PAGE electrophoresis on 10% acrylamide gels [60] and then electroblotted onto nitrocellulose membranes [61]. AKT or phospho-[Ser-473]-AKT was detected following overnight incubation at 4 °C with specific primary antibodies: anti-AKT (1:1000, Cell Signaling, Danvers, MA, USA), anti-phospho-[Ser-473]-AKT (1:1000, Cell Signaling, Danvers, MA, USA). As an internal control, anti-α-tubulin was used. The detection of primary antibody binding employed peroxidase-conjugated secondary antibodies (anti-rabbit IgG, HRP-linked antibody; 1: 1000 Cell Signaling, Danvers, MA, USA) and was visualized utilizing an enhanced chemiluminescent substrate. Chemiluminescent bands were captured using a C-Digit Chemiluminescent Western Blot Scanner (LI-COR, Lincoln, NE, USA), and band intensities were analyzed using LI-COR Image Studio 4.0.

##### Statistical Analysis

The results obtained are described as mean ± standard error of the mean (SEM). One-way ANOVA followed by the Student–Newman–Keuls test were used to compare the intergroup differences and those of the AUC, while paired Student’s *t*-test was used to analyze intragroup differences (*p* < 0.05) using Graphpad Prism^®^ 5.01 (GraphPad Software, San Diego, CA, USA). The data from the rheological studies met the assumptions of normality and homoscedasticity, evaluated by the Shapiro–Wilk and Levene tests, respectively. Thus, the data were analyzed using descriptive statistics followed by two-way independent analysis of variance considering formulations at five levels and temperature at two levels, using the Bonferroni post hoc test for multiple comparisons. The significance level adopted was 0.05 in all analyses, which were performed using Microsoft^®^ Excel^®^ 2010 (Microsoft Corporation) and IBM^®^ SPSS^®^ Statistics version 26.0 (SPSS Inc., Chicago, IL, USA).

## 3. Results and Discussion

The HEC hydrogel was prepared at a concentration of 2% (*w/v*) in Milli-Q water and did not show phase separation during the 30-day storage period, maintaining a homogeneous and translucent aspect. HEC was chosen because it is a nonionic polymer derived from cellulose, soluble in water, and it is widely used as a thickening agent, providing stability to pharmaceutical preparations and as a matrix in controlled release systems, in addition to providing adequate viscosity to the formulations and thus increasing the time of permanence in the nasal mucosa [45,48]. Subsequently, the liposomal dispersion containing insulin functionalized with transduction peptides was added, and the aspect remained homogeneous and translucent [43,62,63,64].

The flow and viscosity curves were obtained by determining the shear stress and viscosity as a function of the shear rate [65]. The rate varied between 0.1 and 100 s^−1^ (upward curve) and between 100 and 0.1 s ^−1^ (downward curve). The total analysis time (upward and downward curves) had a total test time of 120 s, with 24 points taken for each. Rheological flow measurements were carried out in triplicate of formulations containing 2% HEC hydrogel. HEC gel (Natrosol^®^) is formed from the reaction of cellulose with ethylene oxide and has been widely used as a stabilizing agent, a consistency agent for emulsions and a non-ionic gel-forming agent, behaving as a non-Newtonian fluid with a pseudoplastic rheological profile [66]. Analyzing the rheograms (Figure 1A,B), it was observed that the formulations based on the gel composed of 2% HEC, HLI and liposomes functionalized HLI_TAT_ and HLI_PNT_ showed rheological behavior of pseudoplastic fluids (n < 1) (Table 1) at temperatures of 32 °C and 37 °C. It is also noticed that the incorporation of liposomes (HLI, HLI_TAT_ and HLI_PNT_) in hydrogels did not alter the behavior original pseudoplastic characteristic of the HEC hydrogel (pure hydrogel).

Furthermore, according to the results presented in Table 1, the values of the linear regression coefficients (R) obtained from the hydrogels were greater than 0.99 in all formulations analyzed. It should be noted that the temperature did not influence the values of R (*p* > 0.05).

The flow index (n) obtained was less than 1, in accordance with non-Newtonian fluid characteristics with pseudoplastic behavior. According to the statistical evaluation, the flow indexes (n) of HI and HLIpnt were not influenced by temperature, but the other formulations increased with increasing temperature, showing a tendency to transform into Newtonian formulations. On the other hand, the hydrogel and HI formulations obtained the highest flow rates at 32 °C and 37 °C, also showing a tendency towards Newtonian behavior.

Regarding the viscosity measurements, it was verified that when the shear rate is increased, the viscosity values decrease for all the formulations tested, at temperatures of both 32 °C and 37 °C (Figure 1C,D). This type of behavior is characteristic of non-Newtonian fluids with pseudoplastic behavior; a similar profile was observed by Nemen and Lemos-Senna, who tested HEC hydrogels associated with different lipid nanocarriers. In that study, the samples composed of only HEC hydrogels exhibited a reduction in viscosity with an increase in the shear rate [46]. The increase in the shear rate in this case can cause thinning of the flow, known as shear thinning, where it favors intermolecular interactions that cause resistance to flow, making them smaller [65]. In contrast, it is observed that the initial viscosities of all tested formulations (HI, HLI, HLI_TAT_ and HLI_PNT_) were higher when compared to pure hydrogel; this type of behavior was also observed by Chieng and Chen, who found that the inclusion of lipid concentrations in hydrophobically modified HEC hydrogel (HMHEC) increased the viscosity, and this occurred due to hydrophobic bonds aggregating to form micelles, while others were incorporated into the vesicle bilayers, and for this reason, the vesicles could be interconnected, contributing to the increase in viscosity [67]. Additionally, pseudoplastic flow behavior is a desirable property for formulations intended for the administration of drugs via the nasal route, where after shear, the initial resistance for the formulation to flow decreases, reflecting the ease of application [68].

The oscillatory test of the formulations showed that at 32 °C and 37 °C, the samples presented a profile associated with a predominance of G″ in the entire studied range and dependent on the frequency (Figure 1E,F and Table 2), which indicates a predominantly viscous behavior, which can cause an increased absorption of the drug in the nasal cavity due the prolongation of the contact of the formulation, as observed in the study by Mourtas and collaborators [48]. Additionally, a similar rheological behavior was observed in nasal powders developed by Callens and collaborators, who demonstrated an increase in the nasal residence time of insulin in the nasal mucosa [69].

Drug release is an important characterization which was performed to investigate if there were differences between hydrogels, liposomes and functionalized liposomes. The results obtained (Figure 2) demonstrate that after 24 h of testing, 90.74% of insulin was released from the HI formulation, 59.69% from HLI, 42.46% from HLI_TAT_, 41.83% from HLI_PNT_ and ~100% of insulin in solution (IS). Comparing the insulin release profiles, it is observed that after insulin solution, HI was the formulation that released the greatest amount, while the liposomal gel promoted greater drug retention, since the liposomal system provides an extra barrier for the release through the hydrogel. The same behavior was observed by Mourtas and collaborators, where liposomal hydrogels promoted greater yield of the drug compared to its dispersion in simple hydrogel [45]. Additionally, it is observed that both peptides tested had an impact on the insulin release profile, promoting a slower release, due to complexation with insulin. A similar result can be observed in the study by Lin and collaborators, where the liposome functionalization also promoted a prolongation of the triptolide release [70]. Additionally, Shi and collaborators verified that the liposome functionalization with TAT promoted a longer release time of salvianolic acid B, corroborating our findings with more prolonged release caused by peptides [71].

Mathematical modeling describing the release kinetics helps to better understand the in vitro–in vivo correlation of release, employing models such as Weibull, Korsmeyer– Peppas, Baker–Londsdale, First-Order, Hixon–Crowell and Higuchi. According to the determination coefficient r^2^ (Appendix A), we found that the Weibull model best describes the release kinetics of the samples. The Weibull equation relates the amount of drug accumulated as a function of time, and the value of coefficient *b* is an indicator of the mechanism of drug transport through the matrix [54]. For the insulin solution (IS), the coefficient b value was below 0.75, showing a Fickian diffusion mechanism; therefore, the matrix does not control drug release, while for HI, *b* > 1 is observed, indicating that the release of the drug occurred through a complex mechanism that simultaneously involves the relaxation of the polymer chains and the erosion of the polymer. For samples HLI, HLI_TAT_ and HLI_PNT_, values 0.75 > *b* > 1 were observed, demonstrating that the matrix controls drug release, characteristic of the non-Fickian diffusion mechanism.

The nasal insulin administration has been extensively investigated, being considered a promising route of administration for the following reasons: (i) the nasal cavity has an area of highly vascularized epithelial surface available for the absorption of drugs and is covered by numerous microvilli; (ii) the nasal cavity has direct contact with the systemic circulation, preventing the loss of the drug by pre-systemic metabolism in the liver; (iii) possibility of lower doses with rapid achievement of therapeutic blood levels and faster onset of pharmacological effect; (iv) decrease in side effects [10,72,73].

The serum insulin levels (Figure 3) were determined in non-diabetic rats after 60, 120, 180, 240 and 360 min of the intranasal administration of insulin.

Rats receiving nasal saline (SSnasal) had low insulin levels throughout the monitoring period, while rats subcutaneously treated with insulin (INSsub) showed a progressive increase in the insulin serum levels up to 360 min. Rats receiving insulin in saline by intranasal administration (INSnasal) also had a progressive increase in the serum insulin levels, although without homogeneous response, especially at 360 min, suggesting that the nasal insulin absorption was compromised.

Rats receiving HEC-based hydrogels containing insulin (HI) or HEC-based hydrogels + conventional liposome containing insulin (HLI) had improvements in the nasal insulin delivery; in these rats, the serum insulin levels were significantly increased after 60 min, and these levels remained homogeneously high up to 360 min.

Significant increases were observed in the serum insulin levels of rats receiving intranasal administration of HEC-based hydrogels containing liposomes with cell-penetrating peptides (HLITAT and HLIPNT groups). For both groups, insulin serum levels were increased after 60 min of administration, and these levels remained high up to 360 min (*p* < 0.05). The increased insulin levels of the HLITAT group were relatively homogeneous throughout the monitoring period. On the other hand, the HLIPNT group had a significant increase in the insulin levels within 60 min, with a slight decrease occurring at 120–360 min, but these levels were still high in comparison to time 0. These findings with cell-penetrating peptides (CPPs) corroborate previous studies regarding the absorption-promoting effect related to these cationic peptides. On the other hand, the combination of electrostatic interaction and/or complexation of insulin with CPPs and low concentration of CPPs used in this study (0.1 mM) may explain, at least in part, the failure to reach sufficiently high serum insulin levels.

Khafagy et al. studied the effects of PNT and octaarginine CPPs (L and D forms) on nasal insulin release; the authors observed that only the L-form of PNT effectively increased the nasal insulin absorption by increasing insulin permeability without detectable damage to the nasal mucosa. Similarly, studies indicated that L-PNT and the analogue PenetraMax were not systemically or locally toxic at 0.5 mM and 2 mM [34,74]. In another study, Khafagy et al. studied the effect of elimination, substitution and cationicity addition to modified L-PNT residues on nasal insulin absorption and its hypoglycemic effect, validating that insulin was absorbed through the nasal cavity [75].

According to our findings, the administration of the formulations HI, HLI, HLITAT and HLIPNT had satisfactory results regarding the increase in the insulin in vivo disposal. Possibly, the sustained release of insulin may have contributed to this result, as well as the mucoadhesive role of the HEC-based hydrogels, providing a gelled matrix that suggests maintaining a prolonged contact between the drug and the absorption site [39].

D’Souza et al. studied the hypoglycemic effect of insulin intranasally administered into carbopol gel and hydroxypropylmethylcellulose on diabetic animals and on healthy volunteers. The authors observed that the use of this mucoadhesive nasal gel improved the insulin absorption through the nasal mucosa and prolonged the contact between the drug and the absorption sites in the nasal cavity. However, as the absorption was quite rapid, this formulation might not be feasible for long-term chronic use [76].

The glycemia levels (Figure 4) were determined in diabetic rats after 30, 60, 90, 120, 180, 240, 300 and 360 min of the intranasal administration of insulin. In addition, after 360 min of glycemia monitoring, the tibialis anterior muscles were removed to study the changes in the phosphorylation levels of the protein kinase AKT (Figure 5), a component of the insulin signaling involved in the control of various metabolic processes, including the translocation of GLUT-4-containing vesicles to the plasma membrane, favoring glucose uptake [77].

Diabetic rats receiving intranasal administration of saline (SSnasal group) had high glycemia levels throughout the monitoring period. Corroborating these findings, the SSnasal group exhibited low levels of AKT phosphorylation in muscles, suggesting the absence of stimuli to translocate GLUT-4 to the plasma membrane, explaining, at least partially, the increased glycemia levels.

The intranasal administration of HEC-based hydrogels containing empty liposomes (HLB group) in diabetic rats did not improve the glycemia levels and the AKT phosphorylation levels, suggesting that components of the HEC-based hydrogel + empty liposomes (HLB) cannot influence the insulin signaling cascade in tissues.

Diabetic rats treated with insulin subcutaneously (INSsub) had decreases in the glycemia levels soon after 30 min, and these levels practically reached normal levels (considering fasting condition) between 120 and 180 min. After 240 min, glycemia levels began to rise and continued until 360 min. In parallel, these rats had low AKT phosphorylation levels, which is in agreement with the increased glycemia levels in the last 120 min. 

Treatment based on insulin administered by the subcutaneous route may cause patient dissatisfaction. Treatment satisfaction is an important factor in maintaining the treatment adherence and persistence by patients, which is important in the case of diabetes mellitus to prevent long-term complications. Dissatisfaction observed in diabetic patients using subcutaneous insulin administration is mainly associated with injection-related discomfort and risk of hypoglycemia [78,79,80].

Diabetic rats receiving intranasal administration of insulin prepared in saline (INSnasal group) had high glycemia levels throughout the monitoring period. It can be suggested that the insulin absorption efficiency is not favored when administered in saline. Furthermore, the phosphorylation AKT levels were low, in agreement with the high glycemic levels.

Diabetic rats treated with HEC-based hydrogel containing insulin (HI group) or HEC-based hydrogel + conventional liposome containing insulin (HLI group) had a progressive decrease in the glycemic levels throughout the monitoring period. Regarding the diabetic rats from the HLI group, significant decreases in the glycemia levels were observed after 180 min that continued to fall until 360 min, at which time the glycemia levels fell by almost half (241 mg/dL) in relation to the corresponding values before the treatment (463 mg/dL). In the diabetic rats from the HI group, the glycemia levels were significantly decreased after 60 min of treatment; after 180 min, the glycemia levels were practically half (224 mg/dL) of the corresponding values observed at time 0 (460 mg/dL), and remained around 225–230 mg/dL until the end of the monitoring period. For both HI and HLI, it was observed that after a single intranasal administration, there was the gradual drop in the glycemia levels that were maintained low over the 360 min, suggesting that these formulations can be releasing insulin gradually and continuously. The ability of these formulations to progressively decrease the glycemia of diabetic rats is supported by the response profile observed in the temporal evaluation of the serum insulin levels in normal rats (Figure 3). In agreement with these findings, the best results on the AKT phosphorylation levels were observed in diabetic rats from HI and HLI groups: after 360 min of the treatments with HI or with HLI, increased AKT phosphorylation levels were observed in muscles of diabetic rats, suggesting the presence of insulin in the circulation even 360 min after its intranasal administration.

The functionalization of liposomes with CPPs (HLI_TAT_ and HLI_PNT_ groups) modified the ability to change the glycemia levels of diabetic rats. Rats from the HLI_TAT_ group had a progressive decrease in glycemia throughout the monitoring period; however, this glycemic decrease was to a lesser extent than HI and HLI groups. Furthermore, at the end of the 360 min, the glycemia values were about 330 mg/dL, lower than the corresponding values at time 0 (451 mg/dL) but higher than those of HI (230 mg/dL) and HLI (241 mg/dL) groups. The phosphorylation levels of AKT in diabetic rats of the HLI_TAT_ group were slightly increased; however, they were smaller than the increases observed in HI and HLI groups, which corroborates the findings regarding the antihyperglycemic responses for these formulations. Rats from the HLI_PNT_ group had a rapid and progressive drop in the glycemia levels between 60 and 120 min, but after 180 min, the glycemia rose and reached high levels (390 mg/dL) after 360 min. These findings are supported, at least partially, by the response profile observed in the serum insulin levels of normal rats treated with HLI_PNT_—they had significant increases in the insulin levels after 60 min (Figure 3). Taken together, these findings indicate that HLI_PNT_ may be considered a promoter of early insulin intranasal absorption, whereas the HLI_TAT_ may act as a promoter of late absorption. This can be explained, at least partially, by the CPPs + insulin complexation and/or the low concentration of CPPs, below the concentration required for trigger the absorption-promoting effect, as described in the literature [26,28,31].

Attempting to cause a sustained insulin release by using nasal and ocular routes in diabetic rats, Jain et al. developed liposomes coated with chitosan and carbopol for mucoadhesive characteristics. They demonstrated that the insulin release was prolonged for a period of 7 to 9 days, without immediate release (“burst effect”). Conventional liposomes achieved 34% reduction in plasma glucose levels after 8 h, against 35% reduction in 2 days for multivesicular liposomes coated with chitosan [19]. Dyer et al. investigated whether the intranasal insulin delivery by using chitosan could be improved by the application of chitosan-based nanoparticles; the authors observed that chitosan nanoparticles did not improve the effect of increasing insulin absorption when compared with chitosan solution and chitosan powder formulations [56].

## 4. Conclusions

In this study, we report the development and characterization of HEC-based hydrogels containing insulin-loaded liposomes functionalized with cell-penetrating peptides (CPPs), namely TAT and Penetratin (PNT), to improve the nasal absorption of insulin. Rheological measurements were well modeled as non-Newtonian fluids with pseudoplastic behavior, and in vitro release kinetics of insulin from all formulations followed Weibull’s mathematical model. We investigated the enhancement of the insulin biological effects, since these cell-penetrating peptides have been shown to increase the bioavailability of drugs. The results suggest that the intranasal administration of the formulations HI and HLI showed satisfactory and promising results concerning the circulating insulin levels and the antihyperglycemic responses, suggesting that these formulations released insulin gradually and continuously. However, the HLI_TAT_ and HLI_PNT_ formulations have the potential to be improved when compared to the HI and HLI formulations, due to the findings in the literature reporting an excellent performance of these cationic CPPs concerning the ability to promote drug absorption. Therefore, it is suggested that this potential can be enhanced with the use of a higher dose of insulin in liposomes having a combination of TAT and PNT peptides in a single formulation, because according to the results of this study, formulations functionalized with TAT (HLI_TAT_) can be considered promoters of late absorption, and formulations functionalized with PNT (HLI_PNT_) are promoters of early absorption, assuming a treatment carried out concomitantly with subcutaneous insulin administrations, to reduce the number of subcutaneous applications by chronic diabetic patients.

## Figures and Tables

**Figure 1 pharmaceutics-14-02492-f001:**
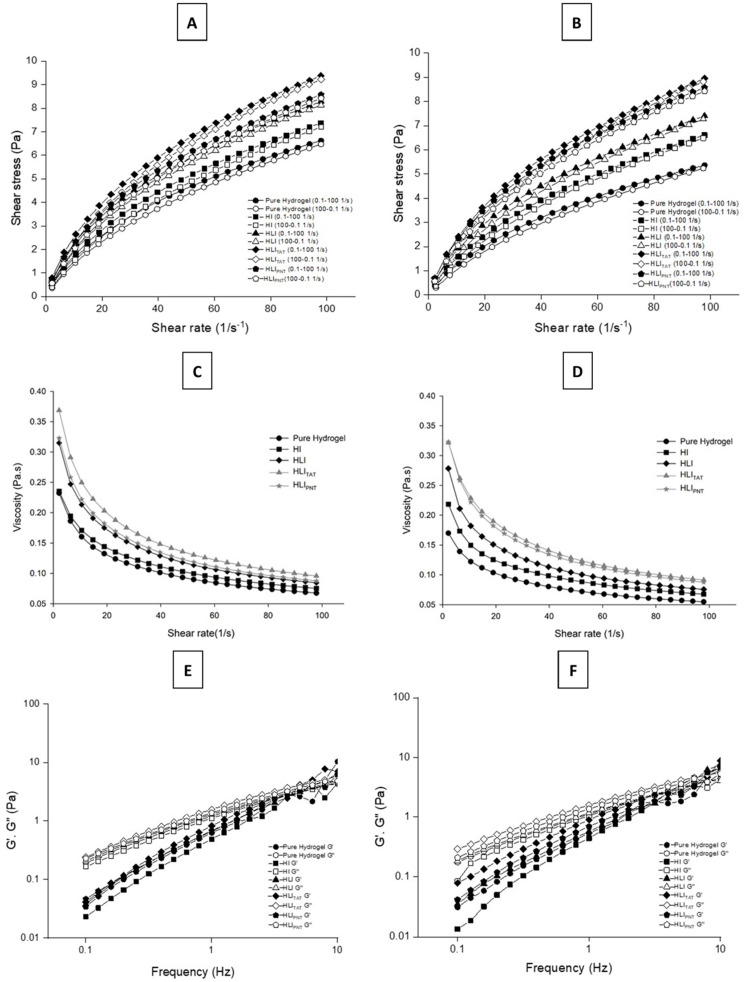
Flow rheograms, viscosity and rheograms of variation in the storage module G′ (filled symbol) and loss G″ (empty symbols) as a function of frequency for all formulations at 32 ± 0.5 °C and 37 ± 0.5 °C (n = 3). For flow rheograms, upward curves have filled symbols and downward curves have empty symbols. Pure HEC-based hydrogel; HEC-based hydrogel containing insulin (HI); HEC-based hydrogel containing insulin-loaded liposomes (HLI); HEC-based hydrogel containing insulin-loaded liposomes functionalized with TAT (HLI_TAT_); HEC-based hydrogel containing insulin-loaded liposomes functionalized with PNT (HLIPNT). (**A**–**D**) Rotational tests; (**E**,**F**) oscillatory measurements. The standard deviations have been omitted for clarity; however, in all cases, the coefficients of variation of the triplicate analyses were less than 5%.

**Figure 2 pharmaceutics-14-02492-f002:**
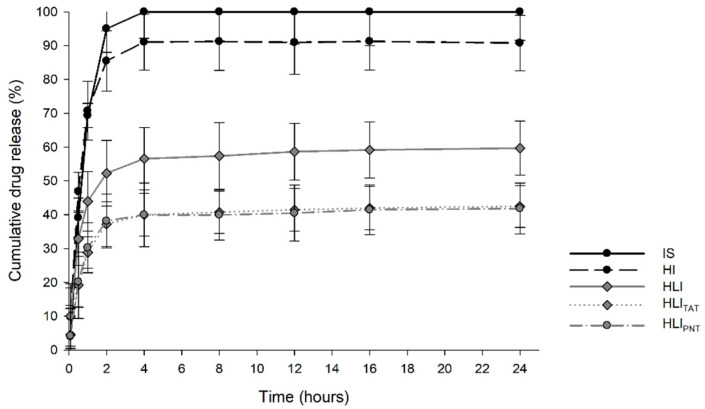
In vitro release profile of insulin by formulations in 24 h. Insulin solution (IS); HEC-based hydrogel containing insulin (HI); HEC-based hydrogel containing insulin-loaded liposomes (HLI); HEC-based hydrogel containing insulin-loaded liposomes functionalized with TAT (HLI_TAT_); HEC-based hydrogel containing insulin-loaded liposomes functionalized with PNT (HLI_PNT_).

**Figure 3 pharmaceutics-14-02492-f003:**
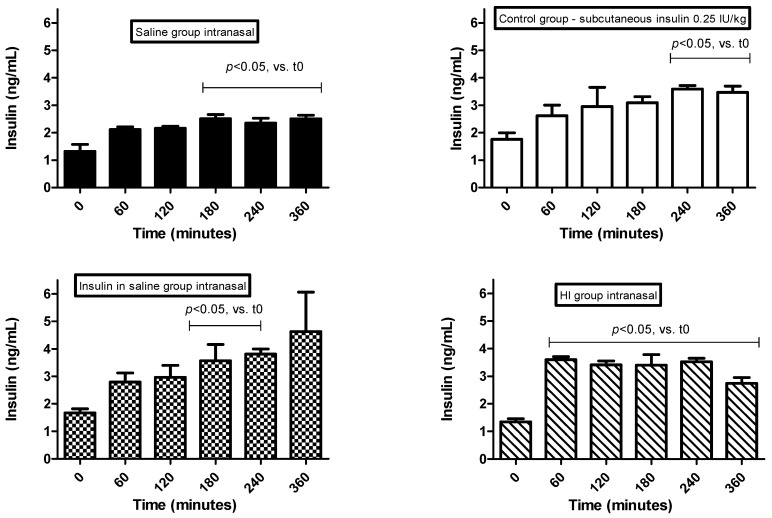
Serum insulin levels (ng mL^−1^) in non-diabetic rats after intranasal administration of insulin. Insulin dose administered subcutaneously: 0.25 IU/animal. Insulin dose administered intranasally: 2.0 IU insulin/formulation/animal. Maximum volume of intranasal instillation: 100 μL divided into 25 μL applications introduced alternately in each nostril. (i) Intranasal administration of saline (0.85% NaCl) (SS_nasal_); (ii) subcutaneous administration of 0.25 IU insulin (Novolin R^®^) (INSsub); (iii) intranasal administration of 2.0 IU insulin prepared in saline (INSnasal); (iv) nasal administration of HEC-based hydrogel containing 2.0 IU insulin (HI); (v) intranasal administration of HEC-based hydrogels containing 2.0 IU insulin-loaded liposomes (HLI); (vi) intranasal administration of HEC-based hydrogels containing 2.0 IU insulin-loaded liposomes functionalized with TAT (HLI_TAT_); (vii) intranasal administration of HEC-based hydrogels containing 2.0 IU insulin-loaded liposomes functionalized with PNT (HLI_PNT_).

**Figure 4 pharmaceutics-14-02492-f004:**
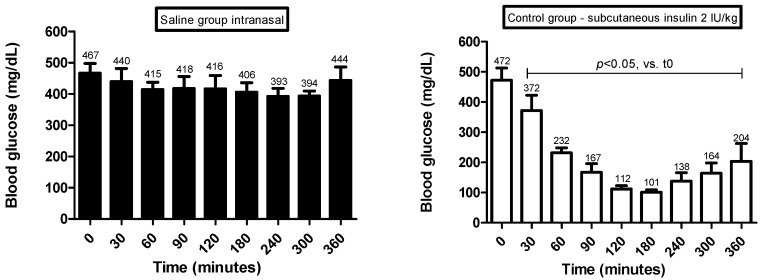
Glycemia levels (mg dL^−1^) in diabetic rats after intranasal administration of insulin. Insulin dose administered by subcutaneous and intranasal: 2.0 IU insulin/formulation/animal. Maximum volume of intranasal instillation: 100 μL divided into 25 μL applications introduced alternately in each nostril. (i) Intranasal administration of saline (0.85% NaCl) (SS_nasal_); (ii) subcutaneous administration of 2.0 IU insulin (Novolin R^®^) (INS_sub_); (iii) intranasal administration of 2.0 IU insulin prepared in saline (INS_nasal_); (iv) intranasal administration of HEC-based hydrogel containing 2.0 IU insulin (HI); (v) intranasal administration of HEC-based hydrogels containing 2.0 IU insulin-loaded liposomes (HLI); (vi) intranasal administration of HEC-based hydrogels containing 2.0 IU insulin-loaded liposomes functionalized with TAT (HLI_TAT_); (vii) intranasal administration of HEC-based hydrogels containing 2.0 IU insulin-loaded liposomes functionalized with PNT (HLI_PNT_); (viii) intranasal administration of HEC-based hydrogels containing blank liposome (HLB).

**Figure 5 pharmaceutics-14-02492-f005:**
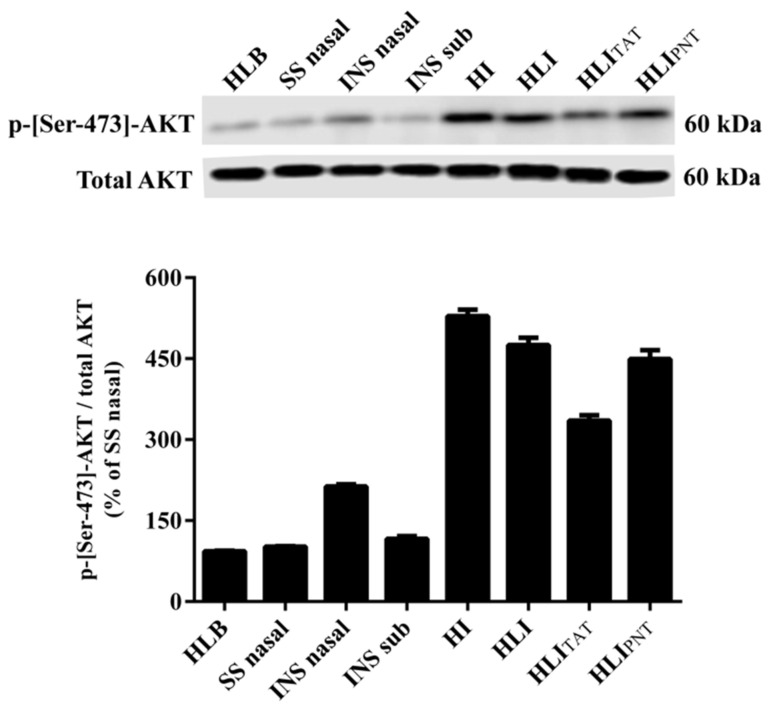
AKT phosphorylation levels (Ser-473) in anterior tibialis muscles of diabetic rats 360 min after intranasal administration of (i) intranasal administration of HEC-based hydrogels containing blank liposome (HLB); (ii) intranasal administration of saline (0.85% NaCl) (SS_nasal_); (iii) intranasal administration of 2.0 IU insulin prepared in saline (INS_nasal_); (iv) control group—subcutaneous administration of 2.0 IU insulin (Novolin R^®^) (INS_sub_); (v) intranasal administration of HEC-based hydrogel containing 2.0 IU insulin (HI); (vi) intranasal administration of HEC-based hydrogels containing 2.0 IU insulin-loaded liposomes (HLI); (vii) intranasal administration of HEC-based hydrogels containing 2.0 IU insulin-loaded liposomes functionalized with TAT (HLI_TAT_); (viii) intranasal administration of HEC-based hydrogels containing 2.0 IU insulin-loaded liposomes functionalized with PNT (HLI_PNT_). The uncropped bolts are shown in Appendix A.

**Table 1 pharmaceutics-14-02492-t001:** Flow index (n) and consistency index (k) and linear regression (R) values at 32 ± 0.5 °C and 37 ± 0.5 °C (n = 3).

Formulations		32 °C			37 °C	
	N	K	R	n	K	R
Pure Hydrogel	0.59146 ^Aa^	0.44693 ^Aa^	0.99785 ^Aa^	0.61356 ^Ab^	0.32128 ^Ab^	0.99791 ^Aa^
HI	0.60650 ^Aa^	0.46455 ^Aa^	0.99768 ^Aa^	0.61916 ^Aa^	0.39254 ^Bb^	0.99829 ^Aa^
HLI	0.56480 ^Ba^	0.62962 ^Ba^	0.99772 ^Aa^	0.58344 ^Bb^	0.51663 ^Cb^	0.99823 ^Aa^
HLI_TAT_	0.55209 ^Ca^	0.75802 ^Ca^	0.99731 ^Aa^	0.56237 ^Cb^	0.68899 ^Db^	0.99670 ^Aa^
HLI_PNT_	0.56144 ^Ba^	0.66301 ^Ba^	0.99737 ^Aa^	0.56329 ^Ba^	0.55264 ^Eb^	0.99789 ^Aa^

Different lowercase letters show a statistical difference between the test temperatures within the same formulation group (*p* < 0.05, Bonferroni). Different capital letters show statistical difference between formulations within the same temperature group (*p* < 0.05, Bonferroni).

**Table 2 pharmaceutics-14-02492-t002:** G′ and G″ values of the oscillatory analysis for the formulations at 32 ± 0.5 °C and 37 ± 0.5 °C (n = 3).

Formulations	32 °C		37 °C	
	G′	G″	G′	G″
Pure Hydrogel	0.61502	1.19306	0.53392	1.11020
HI	0.48396	1.09694	0.44103	1.04699
HLI	0.65501	1.28254	0.57007	1.15608
HLI_TAT_	0.81508	1.55873	0.89033	1.57806
HLI_PNT_	0.67231	1.35616	0.69005	1.36092

## Data Availability

Data are contained within the article.

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
