# Peer review of "Hydroxyethylcellulose-Based Hydrogels Containing Liposomes Functionalized with Cell-Penetrating Peptides for Nasal Delivery of Insulin in the Treatment of Diabetes"

_pharmaceutics, 2022, doi:10.3390/pharmaceutics14112492_

Round 1

Reviewer 1 Report

Blocks of Fig 3 are absolutely blurry. It would be great change the resolution of these graphs. And sholud be indicate that, the ABCD parts came from rotational tests while E and F originated from oscillatory measurements.

The In vitro release profiles should be evaluated by nonlinear regression method. The free of charge "Supplementary material 2" of this article could evaluate the release data: https://doi.org/10.1016/j.molliq.2021.115405

Table 5. should be presented as supplemet material as fig 8 and 9 as well.

Fig 10 is also too blurry. It would be nice change the resolution of this graphs.

Reviewer 2 Report

Authors presenta  new composite medical preparation for the intranasal delivery of insulin to diabetic rats. The study is well planned, the presentation could be more concise.

- the intro should report more references to similar works in which a material and liposomes are combined for drug delivery

- the abstract needs shortening to convey a more concise message

- this expression should be clarified, e.g. Rats were distributed into 10 different groups (n=6) 

- statistical analysis of the data in fig3, for example, is missing and visulaisation could be improved.

Reviewer 3 Report

The manuscript by Chorilli and co-workers describes a platform for nasal delivery of insulin based on hydroxyethylcellulose hydrogels loaded with liposomes functionalized with TAT or PNT peptides. The manuscript should be re-written, it seems like part of a thesis. Some of the data should be in supporting information and some basic aspects of the characterization techniques used should be removed.

The authors should demonstrate unequivocally the advantage of this new delivery system for insulin. Looking at the in vitro results it seems like the amount of insulin released is smaller but like other insulin formulations the release occurs in the first hours after administration. The releasing profile is similar.

The results of the in vivo assays should also be clearly explained since it is not obvious the advantage of using this new platform for insulin.

Round 2

Reviewer 3 Report

The manuscript can now be accepted for publication.